# Research on the Governance Relationship among Stakeholders of Construction Waste Recycling Based on ANP-SNA

**DOI:** 10.3390/ijerph192416864

**Published:** 2022-12-15

**Authors:** Siling Yang, Jie Qiu, Heping Huang

**Affiliations:** 1College of City Construction, Jiangxi Normal University, Nanchang 330022, China; 2School of Civil Engineering and Architecture, East China Jiaotong University, Nanchang 330013, China; 3Institute of Ecological Economics, Jiangxi University of Finance and Economics, Nanchang 330013, China

**Keywords:** construction waste recycling, stakeholder, social network analysis, analytic network process

## Abstract

A method based on Analytic Network Process and Social Network Analysis (ANP-SNA) was proposed in this paper to determine and better clarify the governance relationship among various stakeholders involved. Firstly, fourteen stakeholders of construction waste recycling were identified using the snowball sampling method, and the governance relationships of these stakeholders were summarized into four aspects with eight indicators. Secondly, the weights of the stakeholder governance relationship indicators were determined based on Analytic Network Process (ANP). Thirdly, the Social Network Analysis (SNA) method was used to model the governance relationship network of the stakeholders, and the governance relationships among different stakeholders in the network were described by quantitative analysis of network cohesion, network centrality, structural holes, and other indicators. Finally, key points for optimizing the governance relationships among stakeholders of construction waste recycling were proposed in this paper, so as to provide a new solution for the collaborative governance of stakeholders.

## 1. Introduction

Over the past few years, as urbanization is accelerating, the amount of construction wastes has also been on the rise. According to statistics, construction wastes account for 40% of urban wastes, which poses prominent threats to the ecological environment [1]. Against such a background, more attention has been paid to recycling, the best approach for construction waste disposal. Construction waste recycling involves a large number of stakeholders who interplay with each other, the relationships among whom are complicated and may evolve with the wider application of construction waste recycling, making the stakeholder governance relationships even more complex and uncertain [2]. Therefore, it is necessary to study the governance relationships among the stakeholders of construction waste recycling.

At present, many scholars have already carried out studies on the issue of construction waste recycling stakeholders and the relationships between stakeholders in governance from different perspectives, as evidenced by the following examples. In 1946, Glushge from Russia became the first scholar to propose the concept of construction waste recycling. Yeheyis et al. believed that the governance over construction waste recycling required the participation of various stakeholders [3,4,5]. Ding. et al. explored the modes, mechanisms, and strategies of stakeholders such as the public, enterprises, and social groups participating in construction waste resource management [6]. Melo et al. studied the costs and benefits of stakeholders such as construction waster producers, landfill owners, and resource-based enterprises in disposing construction wastes, and measured them using a cost-benefit approach [7]. As theories concerning construction waste recycling become richer and more mature, the governance relationships among stakeholders have attracted more and more attention of scholars both in China and abroad. Luo et al. suggested that government departments can attract more social capital to participate in the construction waste recycling governance by enhancing the return rate on investment of these projects, and only by doing so can construction waste recycling gain further boosts [8]. Jorge de Brito et al. analyzed the factors affecting the classification and use of construction waste aggregates. The government’s legislation and standardization of construction waste recycled aggregates will help enhance stakeholders’ confidence in the quality of construction waste recycling products, thereby promoting the development of a construction waste recycling industry [9,10,11]. Burcu et al. argued that companies focusing on construction waste recycling could not maximize their benefits without the cooperation between government departments and construction waste producers [12]. Jin et al. pointed out that with the improvement of the ability of construction waste recyclers to recycle waste, construction companies and contractors could enjoy more benefits [13]. So, it is reasonable that the three parties should enhance their cooperation and collaboration. Ma et al. explored the interest relationships among the government, construction waste producers, and resource-based enterprises in construction waste recycling, and constructed a three-party asymmetric evolutionary game model by combining their decision-making behaviors [14].

In a comprehensive view, research on the stakeholders of construction waste recycling has increased in quantity both in China and abroad, and some scholars have also enriched the theories of construction waste recycling from the perspective of stakeholder governance relationship research. However, methods adopted in the existing research are mainly qualitative analysis, cost-benefit analysis, game theory, and so on, and few scholars have used a network method to describe the relationships among the stakeholders and to probe deeply into the interaction between the stakeholders. Various stakeholders are engaged in the governance of construction waste recycling, and their interactions form a relationship network. In this paper, Analytic Network Process (ANP) and Social Network Analysis (SNA) were adopted to conduct a systematic study on the governance relationships among stakeholders of construction waste recycling [15,16,17]. There are two reasons for adopting the method combining ANP and SNA. First, ANP can deal with problems of non-independent hierarchical structure which fail to be addressed by traditional methods in decision-making. Besides, ANP is capable of describing and analyzing the complicated network of governance relationships among stakeholders of construction waste recycling and can determine the weights of relationship indicators. Second, SNA can determine the laws of function of deep networks hidden in complex social systems. When analyzing the interactions and interplay of stakeholders of construction waste recycling, SNA can take a holistic approach to deal with problems in the governance relationships among the stakeholders, thus providing a theoretical basis for the collaborative governance of the stakeholders in construction waste recycling.

## 2. Stakeholders of Construction Waste Recycling and Their Governance Relationships

### 2.1. Stakeholders of Construction Waste Recycling

Stakeholder theory holds that project stakeholders are such individuals or groups that have a certain impact on the realization of project objectives to some extent or will be affected in the process of achieving project objectives [18]. Guided by the stakeholder theory, this paper believes that the individuals or groups who can influence or be influenced by the governance network of construction waste recycling are the stakeholders. Snowball sampling was used to identify stakeholders of construction waste recycling, and the steps are as follows: first, data of construction waste recycling enterprises were collected as the initial samples, and the investigations were conducted on pilot programs for construction waste recycling in cities such as Beijing, Shanghai, Shenzhen, Jinan, Xi’an, and Chongqing, in a bid to select stakeholders; second, experts and scholars in the field of construction waste recycling were invited to make judgments and corrections. As shown in Appendix A, Table A1, a total of 14 stakeholders from four categories were identified.

### 2.2. Governance Relationships among Stakeholders of Construction Waste Recycling

After determining the stakeholders of construction waste recycling, the governance relationships among these stakeholders were identified and optimized by using a literature analysis and expert interviews in this paper. First, keywords were used for the literature retrieval. Through the collation of relevant theoretical literature, it was determined that the governance relationships among stakeholders of construction waste recycling mainly included several aspects such as contract, cooperation, dependence, and communication (these are the first-level indicators) [19,20,21]. Second, these first-level indicators were subdivided into second-level indicators on the basis of suggestions from experts in the interviews to make sure that the research results could be widely and deeply applied.

In order to ensure the feasibility and efficiency of the interview, 10 experts were selected for the semi-structural interviews, including six middle and senior managers of some related parties who had participated in construction waste recycling projects, and four scholars who had many years of research experience in construction waste recycling in universities. The final governance relationship of construction waste recycling stakeholders includes four first-level indicators and eight second-level indicators, as shown in Appendix A, Table A2. Various relationships are interconnected and interdependent, and stakeholders may be correlated to each other in a relationship or in multiple relationships.

## 3. Index Weighting of Governance Relationships among Stakeholders of Construction Waste Recycling

Many existing methods can weight the indicators, such as the correlation coefficient method, analytical hierarchy process, and factor analysis; however, these methods are based on the complete independence of each indicator and ignore the correlations among indicators, which may lead to a distortion of the results. ANP, first proposed by Professor Saaty in 1996, can correlate relationships between indicators at the same level or different levels, and the results of ANP are more accurate and practical [22]. For this reason, ANP was selected in this paper to weight indicators of governance relationships among stakeholders of construction waste recycling.

### 3.1. ANP Introduction

ANP is a decision-making method developed on the basis of analytic hierarchy process (AHP) to adapt to complex structures. ANP builds the relationships between elements in the system into a form similar to the network structure, so as to describe the complex relationships between things in practical problems more accurately.

ANP solves practical problems through the following steps: (1) build control layer and network layer; (2) build judgment matrix: (3) build unweighted super matrix; (4) build weighted super matrix; and (5) calculate limit super matrix.

ANP determines the weight by means of super matrix. However, due to the complexity and difficulty of manual calculation of the super matrix, this paper uses Super Decision software, a special calculation tool of ANP, for the analysis.

### 3.2. ANP Structural Model of Governance Relationships among Stakeholders

According to the characteristics of the governance relationship indicators and comprehensively considering the correlation between the indicators, the ANP model of the governance relationships among stakeholders of construction waste recycling constructed is shown in Figure 1.

### 3.3. Calculation of Index Weights of Governance Relationships among Stakeholders

Since the index system of governance relationships was established, five experts who have been engaged in the research of construction waste recycling and stakeholder governance relationship for many years were invited to discuss the importance of each index according to the 1–9 scale method and determine the final score. In this paper, the judgment matrix was obtained by arithmetic averaging of the same index values, so as to reduce errors caused by experts’ thinking mode and positions.

For example, with the control layer in Figure 1 as the criterion, and element H1 of contract relationship (HT) in the element set of network layer as the sub-criterion, the relative importance of the indicators associated with it in the HT was determined. A judgment matrix was established in the software of Super Decisions, as shown in Figure 2. It can be seen that Inconsistency = 0.0000 < 0.1, which meets the demands of the consistency test.

According to the rule of the above methods, all the index data in the criterion layer were obtained to establish the unweighted super matrix, as shown in Table 1.

With the control layer as the criterion, and elements in the network layer as the sub-criterion, the relative importance between the element sets was judged, so as to obtain the weighted super matrix, as shown in Table 2. The judgment matrix was input using the same steps presented in Figure 2.

Then, the weighted super matrix was self-multiplied to obtain a limit super matrix. Finally, the priorities of the governance relationship index were ranked by the Priorities Command to obtain the comprehensive weights, and the final results are shown in Table 3.

It can be seen from Table 3 that among the second-level indicators of governance relationships among stakeholders in construction waste recycling, indicators such as the validity of contract, resource dependence, and cooperation intention have greater weight values, indicating that indicators have a greater impact on the governance relationship of stakeholders.

## 4. SNA on Governance Relationships among Stakeholders of Construction Waste Recycling

Social network analysis is an analytical method based on sociometrics and graph theory, which uses relational data to quantitatively analyze the characteristics of network structure from multiple perspectives. As one of the most popular social network analysis software, Ucinet software is commonly used for one-dimensional and two-dimensional data analysis.

The stakeholders of construction waste recycling are interconnected, and interact with each other, forming complicated relationships of a network structure. Based on the SNA method, this paper uses Ucinet software to conduct an in-depth study on the structural characteristics of governance relationship network among stakeholders by selecting multiple indicators including network cohesion, network centrality, core edge analysis, and structural holes [23]. The calculation formula of these indicators can be referred to in Appendix B.

### 4.1. Matrix of Stakeholder Governance Relationships

Data of stakeholder governance relationships of construction waste recycling were collected by using a snowball-style questionnaire, and the steps are presented as follows: first, select the construction waste recycling enterprise as the first stakeholder for investigation and then obtain information about other stakeholders relevant to the first stakeholder; second, select an uninvestigated stakeholder from the existing stakeholders for investigation; third, repeat the second step until the investigation objects of all 14 stakeholders mentioned in Section 2.1 are identified and determined. Finally, a total of 500 questionnaires were launched, and 424 questionnaires were collected, with a collection rate of 84.8%. After excluding 61 invalid questionnaires with incomplete answers, identical answers, and inconsistent answers, 363 valid questionnaires in total were obtained, with an efficiency rate of 85.6%.

When establishing a governance relationship network among stakeholders inspired by methods from Almeida et al., the respondents were asked about the direct relationship(s) of their company with stakeholders so as to make clear the structure of relationships among stakeholders of construction waste recycling [24]. Results show that over half of the subjects claimed the presence of such direct relationship(s).

Based on the results of the questionnaire survey, combined with the weights of the indicators shown in Table 3, the already processed basic data of governance relationships among stakeholders were multiplied by the weight of the relationship indicators, so that the governance relationship intensities of various stakeholders can be obtained [25,26,27]. These intensities were then used as weights of edges in the governance relationship network among stakeholders of construction waste recycling to establish the matrix of governance relationships among stakeholders of construction waste recycling. The matrix is shown in Table 4.

### 4.2. Establishment of Relationship Models

In order to facilitate further quantitative analysis on governance relationship network among stakeholders of construction waste recycling, binarization processing was conducted on the relationship matrix in Table 4 by the software of Ucinet according to the conditions that the relationship is marked as “1” if its average intensity value is above 1.51; otherwise, the relationship is marked as “0”. Then, the adjacency matrix was established, and the Ucinet software was used to construct the network model of the governance relationship of stakeholders in construction waste, as shown in Figure 3. Figure values on the graph represent the intensity weights of relationships among stakeholders, and the thick lines indicate strong relationship while slim lines refer to weak relationships.

### 4.3. Analysis on Characteristics of Relationship Network

#### 4.3.1. Analysis on Network Cohesion

In order to analyze the whole cohesion of governance relationship network among stakeholders of construction waste recycling, this paper conducted a network density measurement based on Ucinet software and obtained the stakeholder governance relationship network density [28]. Results show that the network density of this model is 0.5824 and the density standard deviation is 0.4932, indicating the close and strong relationships among stakeholders. That is to say, the change in the behavior of individual stakeholders in the network is more likely to cause behavior changes of other stakeholders in the network.

In SNA, the E-I index was used as a split index to measure how close the relationships between subgroups were. Network density matrices among four category stakeholders were taken as the attribute matrices to measure E-I indexes, and the E-I indexes were then calculated in Ucinet, as shown in Table 5.

It can be seen from Table 5 that the values of E-I indexes of four category stakeholders are all above zero and fall into the range of 0–1, indicating that subgroups of the relationship are relatively open and can communicate with each other.

In terms of the density matrix, construction waste disposal units presented the highest density of 3.724, indicating that the internal network of the group is closely related; construction-related units and government departments ranked second and third, with densities of 3.054 and 2.722, respectively; the public showed the lowest density of 1.048, which showed that the governance relationships among members within the network of stakeholders were not very strong. Besides, in terms of cohesion among the four category stakeholders, the density between construction-related units and construction waste disposal units, and that between construction waste disposal units and government departments are 2.230 and 1.626, respectively, while the density between construction-related units and the public, and that between construction waste disposal units and the public are only 1.017 and 0.870, respectively. These results show that the higher value the density is, the stronger the cohesion strength between the two types of stakeholders is. In general, the network density among the four category stakeholder groups is lower than the network density within each category of stakeholders. This is because relationships among different category of stakeholders are weak while relationships within each stakeholder group are strong.

#### 4.3.2. Analysis on Network Centrality

The network centrality of stakeholders was analyzed by Multiple Measures in Ucinet, and the results are shown in Table 6.

It can be observed from Table 6 that the degree centrality and closeness centrality of the construction administration department are far ahead of other stakeholders, so these departments are at the absolute center; followed by the construction unit and recycling enterprise in the sub-center; the point centrality value and closeness centrality value of media, transportation unit and environmental assessment agency are the lowest. These results indicate that the construction administration department, as the core node, has the greatest influence in the network, and exerts influence on other stakeholders to a large extent; the four stakeholders, including construction administration department construction unit, construction company, and recycling enterprise, are located in the center of the whole network, because they frequently exchange information and are less restricted by other stakeholders within the network [29,30]. In contrast, the media, transportation unit, and environmental assessment agency have weak influence within the network and are prone to be affected by, and dependent on, other stakeholders to a large extent.

The intermediate centrality values of the four stakeholders of the construction administration department, the construction company, the construction unit, and the recycling enterprise are in the leading position, whose intermediate centrality values are 21.987, 5.021, 4.167 and 4.167, respectively, while the intermediate centrality values of other eight stakeholders are all below 4. These results have two indications. First, the front four stakeholders serve as the distribution center of network information interactions in the whole network. These four stakeholders are the bridge of information interactions among different stakeholders within the network, controlling the information resources of the whole network. Second, within the whole network, only these four stakeholders have effective intermediate centrality values, which indicate there is a lack of effective network information interaction among other stakeholders.

#### 4.3.3. Core-Periphery Analysis

In Ucinet, core-periphery analysis was conducted on the governance relationship network among stakeholders of construction waste recycling according to the menu path of Network-Core/Periphery-Categorical. The results are shown in Table 7.

It can be seen from Table 7 that with the core degree of 0.3 as the boundary, the core and peripheral stakeholders of construction waste recycling are distinguished. The construction administration department construction unit, recycling enterprise, construction company, disposal site, and municipal environmental sanitation administration department are classified as “core zone”, and the other eight stakeholders as “peripheral zone”. The average core degree of the core zone is 0.337 and that of the peripheral zone is 0.187, indicating that these two zones have obvious differences in relationship structure.

According to Table 4 and Table 7, the average value of the relationship strength of the whole relationship network is 1.51. In the core zone, average values of strength of relationships among stakeholders are all above 1.51, of which the highest is 2.304 for the construction company, the lowest is 1.673 for the municipal environmental sanitation administration department, and the average value is 2.029. In the peripheral zone, relationship strength among the other eight stakeholders is relatively low, with an average value of 1.347. These figures indicate that the relationship strength of the core zone is much higher than that of the peripheral zone; the stronger the relationship strength, the more core is the position of the stakeholder in the network [31].

#### 4.3.4. Analysis on Structural Holes

Analysis on structural holes can effectively distinguish between the different structural characteristics of relationship subjects in the network. Based on the specific measurement method of structural holes theory, the measurement data results of the network structure hole of governance relationships among stakeholders of construction waste recycling were calculated by Ucinet software and are shown in Table 8.

It can be seen from Table 8 that the construction administration department has the highest efficient scale and occupies a dominant position in the network, so it has strong control over the interactions of other stakeholders. The construction administration department, media, and construction company achieve higher scores in efficiency degree, indicating that these three stakeholders conduct high-efficiency actions in the network. The construction administration department, construction company, construction unit, and recycling enterprise have small restriction degrees, indicating that these four stakeholders are located in the structural hole position in the network and control the core resources. However, the media, transportation unit, and environmental assessment agency have weak capability in utilizing structural holes, so these three stakeholders are somewhat “dissociated” with other stakeholders.

## 5. Conclusions

Based on the stakeholder theory, this paper established a governance relationship network among stakeholders of construction waste recycling by using ANP-SNA, and analyzed the constructed network model. Results showed that the network can be optimized by considering the following aspects:

(1)In terms of network cohesion, various stakeholders have close contacts with each other, and there is no obvious clique in the relationship network, but the collaborative activities of stakeholders tend to be carried out within each category of stakeholders, and the collaboration between various categories of stakeholders in the network needs to be strengthened so as to facilitate the integration and coordination capability of the network;(2)As the core of the whole network, the construction administration department is controlling the whole network, influencing, and even determining, the relationships between other stakeholders in the network. For this reason, the construction administration department, which has the greatest influence in the network, should give full play to its role as the core of the network, and maximize the efficiency under the premise of satisfying the interests of all stakeholders so as to realize collaborative governance of stakeholders for construction waste recycling;(3)The members of the core zone, such as the construction administration department and construction unit have a strong network relationship strength, as they maintain frequent contacts and information exchanges with other stakeholders. In contrast, members of the peripheral zone, such as the media and environmental assessment agency, obviously are overly marginalized rather than fully involved, so they fail to give full play to their potential as stakeholders. Therefore, the functions of these peripheral stakeholders should be enhanced;(4)The three stakeholders including construction administration department, media, and construction company have the highest influence efficiency in the whole network, indicating that they can influence the cooperation between other stakeholders in a more efficient and direct way. Four stakeholders, including construction administration department, construction company, construction unit, and recycling enterprise are located in places with rich structural holes, so they can improve the cooperation efficiency of the whole network by guiding the collaborative cooperation among various stakeholders;(5)Overall, to realize collaborative governance of stakeholders for construction waste recycling, the once separated subjects must be integrated and linked into a whole so that all relevant stakeholders can work together to achieve the balanced development of interests and optimal governance.

This paper proposes a visualized and innovative research method based on ANP-SNA, which can solve the problem of interplay and interaction among stakeholders of construction waste recycling and can accurately quantify the correlations of governance relationships among stakeholders. However, this paper also has the following deficiencies. First, this paper assumes that the governance relationships among stakeholders are undirected and fails to take into consideration the direction of information and resource transmission. Second, in the long term, stakeholders of construction waste recycling are changing both in their roles and positions, so in the future more emphasis should be laid on how to effectively analyze the dynamic evolution of the relationship network.

## Figures and Tables

**Figure 1 ijerph-19-16864-f001:**
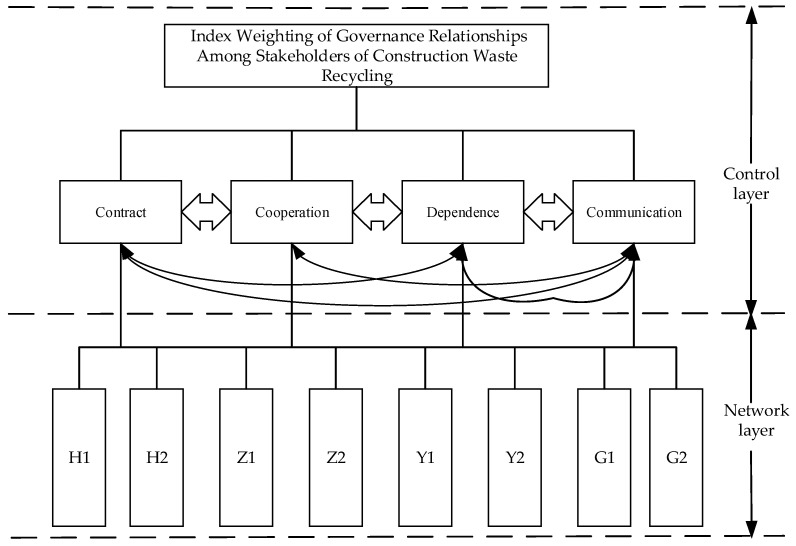
ANP model of governance relationships among stakeholders.

**Figure 2 ijerph-19-16864-f002:**
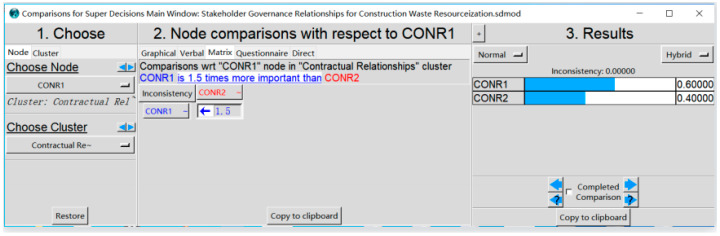
Pairwise comparison matrix of elements in HT set under H1 criterion.

**Figure 3 ijerph-19-16864-f003:**
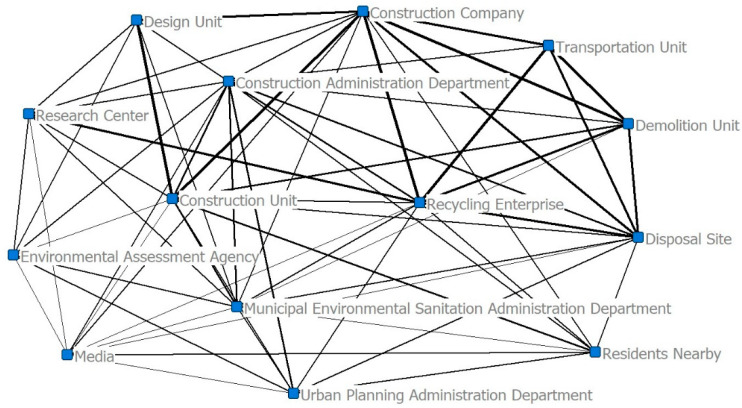
Model graph of governance relationship network among stakeholders.

**Table 1 ijerph-19-16864-t001:** Unweighted super matrix.

Index	H1	H2	Z1	Z2	Y1	Y2	G1	G2
H1	0.600	0.400	0.615	0.428	0.532	0.428	0.444	0.453
H2	0.400	0.600	0.384	0.571	0.467	0.571	0.555	0.546
Z1	0.571	0.461	0.428	0.307	0.384	0.301	0.307	0.357
Z2	0.428	0.538	0.571	0.692	0.615	0.699	0.692	0.642
Y1	0.539	0.444	0.428	0.532	0.666	0.222	0.460	0.333
Y2	0.460	0.555	0.571	0.467	0.333	0.777	0.539	0.666
G1	0.444	0.428	0.375	0.500	0.400	0.400	0.724	0.272
G2	0.555	0.571	0.625	0.500	0.600	0.600	0.275	0.727

**Table 2 ijerph-19-16864-t002:** Weighted super matrix.

Index	H1	H2	Z1	Z2	Y1	Y2	G1	G2
H1	0.200	0.133	0.153	0.107	0.149	0.120	0.126	0.129
H2	0.133	0.200	0.096	0.142	0.131	0.160	0.158	0.156
Z1	0.126	0.102	0.136	0.097	0.092	0.072	0.043	0.051
Z2	0.095	0.119	0.181	0.220	0.147	0.167	0.098	0.091
Y1	0.139	0.115	0.107	0.133	0.213	0.071	0.098	0.071
Y2	0.119	0.144	0.142	0.116	0.106	0.248	0.115	0.142
G1	0.082	0.079	0.068	0.090	0.064	0.064	0.258	0.097
G2	0.102	0.105	0.113	0.090	0.096	0.096	0.098	0.259

**Table 3 ijerph-19-16864-t003:** Index weights of governance relationships.

Second-Level Indicator	Global Weight
Timeliness of the contract (H1)	0.140
Efficiency of the contract (H2)	0.150
Shared goals (Z1)	0.091
Cooperation intention (Z2)	0.141
Time dependence (Y1)	0.119
Resource dependence (Y2)	0.144
Communication frequency (G1)	0.095
Communication effect (G2)	0.120

**Table 4 ijerph-19-16864-t004:** Matrix of governance relationships among stakeholders.

Number	S1	S2	S3	S4	S5	S6	S7	S8	S9	S10	S11	S12	S13	S14
S1	0	3.075	2.934	2.467	1.806	1.865	1.662	2.884	2.743	1.803	2.038	2.217	1.894	1.542
S2	3.075	0	2.157	2.449	1.801	2.037	1.095	1.942	1.851	0	1.356	2.157	2.062	1.447
S3	2.934	2.157	0	2.066	0	0	0	1.82	1.588	0	1.679	0	1.854	1.215
S4	2.467	2.449	2.066	0	3.96	4.031	2.666	2.232	2	0	2.451	1.896	1.478	1.356
S5	1.806	1.801	0	3.96	0	3.666	0	0	0	0	0	1.851	1.71	0
S6	1.865	2.037	0	4.031	3.666	0	4	4.055	3.431	3.648	1.942	2.086	0	1.5
S7	1.662	1.095	0	2.666	0	4	0	3.816	3.553	4.026	0	0	0	0
S8	2.884	1.942	1.82	2.232	0	4.055	3.816	0	3.235	4.29	2.037	3.345	0	1.359
S9	2.743	1.851	1.588	2	0	3.431	3.553	3.235	0	3.648	2.276	0	0	1.43
S10	1.803	0	0	0	0	3.648	4.026	4.29	3.648	0	0	0	0	0
S11	2.038	1.356	1.679	2.451	0	1.942	0	2.037	2.276	0	0	0	0	1.989
S12	2.217	2.157	0	1.896	1.851	2.086	0	3.345	0	0	0	0	1.823	1.261
S13	1.894	2.062	1.854	1.478	1.71	0	0	0	0	0	0	1.823	0	1.215
S14	1.542	1.447	1.215	1.356	0	1.5	0	1.359	1.43	0	1.989	1.261	1.215	0

**Table 5 ijerph-19-16864-t005:** Density matrices and E-I indices among stakeholders.

Density Matrix	Government Departments	Construction-Related Units	Construction Waste Disposal Units	The Public	E-I Index among Stakeholders
Government Departments	2.722	1.437	1.626	1.622	0.333
Construction-related Units	1.437	3.054	2.230	1.017	0.610
Construction Waste Disposal Units	1.626	2.230	3.724	0.870	0.333
The Public	1.622	1.017	0.870	1.048	0.520

**Table 6 ijerph-19-16864-t006:** Analysis on Network Centrality.

Stakeholder	Centrality			
	Point	Closeness	Between	Feature Vector
Construction Administration Department	100.000	100.000	21.987	53.789
Municipal Environmental Sanitation Administration Department	69.231	76.471	2.949	42.918
Urban Planning Administration Department	53.846	68.421	1.859	34.726
Construction Unit	76.923	81.250	4.167	47.382
Design Unit	46.154	65.000	0.577	29.599
Construction Company	76.923	81.250	5.021	46.503
Demolition Unit	46.154	65.000	0.256	31.511
Recycling Enterprise	76.923	81.250	4.167	47.049
Disposal Site	69.231	76.471	2.585	43.366
Transportation Unit	38.462	61.905	0.000	26.468
Residents Nearby	53.846	68.421	3.419	33.737
Research Center	53.846	68.421	1.154	34.606
Environmental Assessment Agency	38.462	61.905	0.577	23.302
Media	15.385	54.167	0.000	10.425

**Table 7 ijerph-19-16864-t007:** Core Degree of Stakeholders.

Number	Stakeholder	Core Degree	Ranking
1	Construction Administration Department	0.404	1
2	Municipal Environmental Sanitation Administration Department	0.302	6
3	Urban Planning Administration Department	0.237	7
4	Construction Unit	0.341	2
5	Design Unit	0.198	11
6	Construction Company	0.333	4
7	Demolition Unit	0.215	10
8	Recycling Enterprise	0.339	3
9	Disposal Site	0.307	5
10	Transportation Unit	0.178	12
11	Residents Nearby	0.231	9
12	Research Center	0.236	8
13	Environmental Assessment Agency	0.152	13
14	Media	0.069	14

**Table 8 ijerph-19-16864-t008:** Analysis on Structural Holes of Stakeholders.

Organization	Efficient Scale	Efficiency Degree	Restriction Degree
Construction Administration Department	6.846	0.527	0.266
Municipal Environmental Sanitation Administration Department	3.222	0.358	0.384
Urban Planning Administration Department	2.429	0.347	0.476
Construction Unit	3.800	0.380	0.353
Design Unit	1.667	0.278	0.551
Construction Company	4.000	0.400	0.352
Demolition Unit	1.333	0.222	0.555
Recycling Enterprise	3.800	0.380	0.353
Disposal Site	3.000	0.333	0.388
Transportation Unit	1.000	0.200	0.648
Residents Nearby	2.714	0.388	0.470
Research Center	2.143	0.306	0.482
Environmental Assessment Agency	1.800	0.360	0.627
Media	1.000	0.500	1.125

## Data Availability

Data are available from the authors upon request.

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
