# Peer review of "Research on the Governance Relationship among Stakeholders of Construction Waste Recycling Based on ANP-SNA"

_ijerph, 2022, doi:10.3390/ijerph192416864_

Round 1
Reviewer 1 Report
The authors assessed the governance relationship among stakeholders of construction waste recycling based on Analytic Network Process (ANP) and Social Network Analysis (SNA). The topic is interesting and the article is well structured. Some minor comments are presented:
- "ANP-SNA" should be defined on its first appearance in the abstract.
- I don't see any reference from Prof. Jorge de Brito, which is the most recognized researcher in the field of construction waste recycled aggregate. Please find some suggestions below, which I believe can improve the robustness of the Introduction:
doi:10.1016/j.cemconcomp.2004.07.005
doi.org/10.1016/j.jobe.2021.102187
doi.org/10.1016/j.jclepro.2016.12.070
- The conclusion section may be summarized. The two paragraphs below bullet (4) should be shortened.
Reviewer 2 Report
A method based on ANP-SNA is proposed in this paper to figure out and better clarify 12 the governance relationship among various stakeholders involved, and proposed new solutions. The article needs to be revised before it is accepted.
1. The content in Figure 3 is not clear and cannot be recognized.
2. There are too many tables in the text, and it is a better choice to put them in the support material.
3. The statement in the conclusion needs to be refined.
Reviewer 3 Report
This paper presents a study to model, based on Analytic Network Process and Social Network Analysis, the governance interrelationship between the stakeholders involved in Construction Waste Recycling. By better understanding how these stakeholders interact, it is possible to optimise the governance relationships among stakeholders of construction waste recycling.
The study is interesting but not very well explained. The introduction is practical and straightforward. I think the methodology is well structured but not well described. It should be explained more clearly how the factors and weights that describe the relationship between the indicators and elements of the system are calculated. There is a problem with the wording and the use of English. Also, the use of specific software is often taken for granted and is not adequately described. The conclusions are interesting, and I believe this is the study's most relevant part.
The topic covered is of utmost importance to reduce the human environmental footprint. It is necessary to systematise construction waste management to treat it as a renewable resource. I believe, however, that the methodology should be better described and in greater detail so that other researchers can reproduce it.
-
Lines 49 and 63: This document will be read by researchers from different countries. It is unclear what context the authors are referring to when the word "home" is used. Please be more specific as to which country is being referred to.
-
Paragraph between lines 138 and 143 not understandable.
-
Line 183: incomplete reference, missing year.
Round 2
Reviewer 3 Report
The article is much improved. The material in the appendices can be of great use to other researchers in the field.